# Near-Real-Time Tephra Fallout Assessment at Mt. Etna, Italy

**Simona Scollo** [1],*, **Michele Prestifilippo** [1], **Costanza Bonadonna** [2], **Raffaello Cioni** [3],
**Stefano Corradini** [4], **Wim Degruyter** [5], **Eduardo Rossi** [2], **Malvina Silvestri** [4], **Emilio Biale** [1],
**Giuseppe Carparelli** [3], **Carmelo Cassisi** [1], **Luca Merucci** [4], **Massimo Musacchio** [4] and
**Emilio Pecora** [1]

[1] Istituto Nazionale di Geofisica e Vulcanologia, Osservatorio Etneo, 95125 Catania, Italy;
    michele.prestifilippo@ingv.it (M.P.); emilio.biale@ingv.it (E.B.); carmelo.cassisi@ingv.it (C.C.);
    emilio.pecora@ingv.it (E.P.)
[2] Département des sciences de la Terre, Université de Genève, 1205 Geneva, Switzerland;
    Costanza.Bonadonna@unige.ch (C.B.); Eduardo.Rossi@unige.ch (E.R.)
[3] Dipartimento di Scienze della Terra, 50121 Firenze, Italy; raffaello.cioni@unifi.it (R.C.);
    giuseppe.carparelli@stud.unifi.it (G.C.)
[4] Istituto Nazionale di Geofisica e Vulcanologia, 00143 Roma, Italy; stefano.corradini@ingv.it (S.C.);
    malvina.silvestri@ingv.it (M.S.); luca.merucci@ingv.it (L.M.); massimo.musacchio@ingv.it (M.M.)
[5] School of Earth and Ocean Sciences, Cardiff University, Cardiff CF10 3AT, UK; DegruyterW@cardiff.ac.uk
*   Correspondence: simona.scollo@ingv.it; Tel.: +39-095-7165858

**Abstract:** During explosive eruptions, emergency responders and government agencies need to make
fast decisions that should be based on an accurate forecast of tephra dispersal and assessment of the
expected impact. Here, we propose a new operational tephra fallout monitoring and forecasting
system based on quantitative volcanological observations and modelling. The new system runs at
the Istituto Nazionale di Geofisica e Vulcanologia, Osservatorio Etneo (INGV-OE) and is able to
provide a reliable hazard assessment to the National Department of Civil Protection (DPC) during
explosive eruptions. The new operational system combines data from low-cost calibrated visible
cameras and satellite images to estimate the variation of column height with time and model volcanic
plume and fallout in near-real-time (NRT). The new system has three main objectives: (i) to determine
column height in NRT using multiple sensors (calibrated cameras and satellite images); (ii) to compute
isomass and isopleth maps of tephra deposits in NRT; (iii) to help the DPC to best select the eruption
scenarios run daily by INGV-OE every three hours. A particular novel feature of the new system
is the computation of an isopleth map, which helps to identify the region of sedimentation of large
clasts (≥5 cm) that could cause injuries to tourists, hikers, guides, and scientists, as well as damage
buildings in the proximity of the summit craters. The proposed system could be easily adapted to
other volcano observatories worldwide.

**Keywords:** hazard assessment; column height; near-real-time forecasts; maximum clast forecasts;
operational system; Etna volcano

## 1. Introduction

### 1.1. Worldwide Operational Monitoring and Forecasting of Tephra Dispersal and Fallout

Tephra dispersal and fallout represent a source of multiple hazards and impacts for society,
including health issues for humans [1] and animals [2], and damage to residential buildings and
infrastructure [3], transportation systems [4], and agriculture [5–7]. While the fallout of ash and small to

medium lapilli has been widely considered as a primary risk agent related to explosive volcanic activity, fallout of coarse lapilli to small blocks falling from plume margins has been underrated. As an example, during the event at Etna on 23 November 2013, clasts from several centimeters to decimeters fell within 5–6 km from the summit and hit hikers who were in the touristic areas [8]. Although the assessment of tephra fallout and dispersal in distal areas has been largely considered [9–12], the reduction of volcanic impacts in proximal areas and within the first hour from the beginning of the eruption is still a challenge. As a matter of fact, regardless of the importance of this information for emergency responders and government agencies, the operational systems capable of monitoring tephra dispersal and fallout in near-real-time (NRT) and returning the expected impact assessment are still limited and not fully adapted to the growing requirements of precision and reliability.

A good example of NRT tephra detection in volcano observatories is represented by the Alaska Volcano Observatory (AVO), which monitors volcanoes within the North Pacific region [13]. The AVO system analyzes data from different satellite sensors. They use a 24/7 automated ash cloud detection algorithm that sends emails and phone text alerts to the AVO members, who are, in turn, responsible for verifying if the automatic alert can be considered as true or false [13]. The Kamchatka Volcanic Eruption Response Team (KVERT) monitors 36 active volcanoes in Kamchatka and the Kuril Islands and analyzes real-time satellite data daily in cooperation with the Institute of Volcanology and Seismology of the Far East Branch of the Russian Academy of Sciences [14,15]. They have a system that integrates data from different satellite platforms with meteorological data and land-based information [15]. If data include the onset of an eruption, simulations using the model PUFF are automatically run to assess column height and azimuth of dispersal of the ash plume [15]. At the Iceland Met Office (IMO), volcanic ash plumes are continuously monitored by different instruments as C-band weather radars and X-band mobile radars. Forecasts using two different models named NAME [16] and VOL-CALPUFF [17] are run in the case of an eruption in several volcanoes, using column heights from radar data. Japanese volcanoes are also monitored using satellite systems by the Japan Meteorological Agency (JMA) and the Meteorological Satellite Center (MSC), which provides information on volcanic ash plumes to the Tokyo Volcanic Ash Advisor Center (VAAC). Moreover, the JMA produces preliminary ash fall forecasts 5–10 min after an eruption and full forecasts every 20–30 min based on observational reports of the ongoing eruption (e.g., column height, duration) obtained using different instruments [18,19]. At Etna volcano, since 2006 a tephra monitoring and forecasting system has been fully working at the Istituto Nazionale di Geofisica e Vulcanologia, Osservatorio Etneo (INGV-OE) [20]. While observations of explosive activity are done with different instruments, the tephra fallout forecasts are based on models that are run with fixed eruptive scenarios whose input parameters are associated with past eruptions [20–23]. These scenarios, named strong plume scenarios (SPS) and weak plume scenarios (WPS), represent two end members of Etna eruptions. Maps of tephra fallout forecasts are automatically sent to the DPC, who make them available to the local authorities.

Operational tephra dispersal and sedimentation models, such as the FLEXPART model [24], the PUFF model [25], the HYSPLIT model [26], the TEPHRA model [5], and the FALL3D model [27], are now widely used. However, it is well-known that the reduction of uncertainty in model outcomes cannot be done without the integration of observational data coming from an advanced monitoring system [28,29]. The new challenge of modern volcanology is, hence, the continuous improvement of data assimilation, which includes the use of data from field or remote sensing systems to solve inverse problems that better characterize the eruption source parameters (ESPs). Examples may include variational data assimilation methods [30] or Bayesian approaches [31]. A more direct approach is to derive ESPs by analyzing data obtained from ground or satellite remote sensing systems, thereby greatly increasing the model accuracy [32–34].

## 1.2. Main Objectifies of the New Upgraded System at INGV-OE

In the last twenty years, more than one hundred explosive events have generated high eruption columns (2–15 km) and copious tephra fallout that entirely covered the flanks of Etna volcano

(~$10^4$–$10^6$ m$^3$) [35,36]. Most of the events that occurred after 2011 had different features and the mass of the erupted tephra changed from one event to another [37]. ESPs for those events could significantly differ from ESPs used in the WPS and SPS sent to the DPC. Moreover, after the event on 23 November 2013, the need emerged to identify the area covered by large clasts falling from plume margins that could reach touristic paths. As a consequence of the significant hazard associated with those events, the DPC funded a specific project in order to: (i) best select results of simulations computed using ESPs of WPS or SPS; (ii) run a simulation for tephra deposits using data acquired in NRT; (iii) define the area which, in case of a lava-fountain-fed plume, would be covered by fallout of large clasts that could cause injuries to tourists, hikers, and scientists, and damage infrastructure in proximal areas. In this paper, we present the new system operating at INGV-OE that is able to perform and send to the DPC tephra deposit simulations in NRT (after several hours from the beginning of the eruptive event) based on parameters measured using observations collected in real-time.

## 2. Materials and Methods of the New Monitoring and Forecasting Structure

### 2.1. Monitoring Structure

The monitoring of Etna volcanic plumes utilizes data from two visible cameras and the spinning enhanced visible and infrared imager (SEVIRI) satellite. The plume height is estimated in NRT by volcanologists by making use of both cameras and satellite images. The new camera network consists of two low-cost, high definition visible cameras. The model of the cameras is VIVOTEK IP8172P, with a 1/2.5″ progressive CMOS (complementary metal oxide semiconductor) image sensor and a maximum resolution of 2560 × 1920. The focal length is 3.3–10.5 mm and the aperture is F 1.6 (wide)–F 2.7 (tele). The field-of-view is 33°–93° (vertical), 24°–68° (horizontal), and 40°–119° (diagonal). The locations of the cameras are shown in Figure 1. One sits on the southern flank of Etna in Catania (Etna Cuad Visible High definition camera (ECVH): 15.0435°; 37.5138°; site A in Figure 1) and the other one on the west flank of Etna near Bronte town (Etna Bronte Visible High definition camera (EBVH): 14.8568°; 37.8096°; site B in Figure 1). The settings for the EBVH camera are horizontal field of view (HFOV) 26.2, vertical field of view (VFOV) 34.5, and field of view (FOV) 42.5. For each of the cameras, we proceeded as follows: (1) the camera was calibrated as described in [35] (the positions of the cameras are well-known using the Global Positioning System (GPS) location); (2) the estimation of the orientation was made using a 3D tool aimed to simulate the cameras and was by performed by aligning the skyline with the digital elevation model (DEM) of Etna volcano; (3) to estimate the plume height, we assumed that the plume had a negligible depth and that it was confined to a vertical plane that rotated according to the wind direction. The uncertainty of the column height was set +/− 500 m and mainly depended on the distance among the horizontal lines drawn in the calibrated visible cameras shown in Figure 2a,b. The calibrated images of both cameras were available to the INGV-OE volcanologists and DPC through a dedicated website and via folders with restricted access.

Since 2010, a station installed in Rome has acquired images from Meteosat Second Generation (MSG). The MSG satellites carry the spinning enhanced visible and infrared imager (SEVIRI), which observes the Earth in 12 spectral channels: four visible and near-infrared (VNIR) channels, and eight infrared (IR) channels. The imaging sampling distance is 3 km at the subsatellite point for standard channels, and 1 km for the high-resolution visible (HRV) channel. The station integrates hardware and software for automated reception of data and preprocessing of SEVIRI data from the Meteosat satellite, providing imagery every 5–15 min over Italy. The combination of the high temporal resolution and spectral capabilities makes these data relevant for the monitoring of volcanic activity [38,39]. The data acquired in native format are automatically calibrated and converted in radiance and saved in hierarchical data format (HDF). HDF data are then stored in a dedicated directory and sent by file transfer protocol (FTP) to different directories. The column height during an eruption is estimated using the comparison between the brightness temperature of the most opaque pixel with the atmospheric temperature profile extracted from the weather forecasts around the volcano [38–40]. The error is

estimated by considering a brightness temperature uncertainty of +/−2 K, which for Etna means an uncertainty that is generally less than 10% in height (in general +/−500 m for satellite data). In our automatic system, we consider a square centered at the summit crater with measuring 9 pixels (about 27 km) on each side. An automatic software downloads and archives SEVIRI data that are also visualized in the INGV-OE monitoring room and is a twenty-four seven service (24/7). Another semiautomatic software is available to the volcanologists, who can read the estimated column height value by simply clicking the mouse above the plume identified on the image. Note that the described procedure for the column height retrieval is extremely easy, but is only reliable if the "darkest pixel" can be considered as completely opaque (i.e., the contribution to the top-of-atmosphere (TOA) radiance measured from the satellite coming from the surface or the atmosphere under the cloud is negligible. Such a condition is difficult to obtain, and this leads to a general underestimation of the satellite height retrievals. A correction for this effect is under development. Images from camera and satellite sensors visible to the volcanologists are shown in Figure 2. It is worth noting that the location of the volcanic vent does not impact the column height estimations using both camera and satellite sensors. Components and lower and upper limit costs, disadvantages, and alternative solutions of the proposed monitoring systems are described in Table 1.

**Table 1.** Details of the lower cost limits, components, disadvantages, and possible solutions.

| Sensors | Lower Cost Limit | Components | Disadvantages | Solutions |
|---|---|---|---|---|
| Cameras | ~3000 € | Two visible cameras, acquisition system | No measurements during the night | Use of thermal cameras with an increase of the costs |
| SEVIRI satellite | ~3000 € | Acquisition system, parabolic antenna and signal reception, open source acquisition software | Greater errors are expected when the volcanic plume is not of optical thickness | Data taken from external platforms acquired in near-real-time (NRT) |

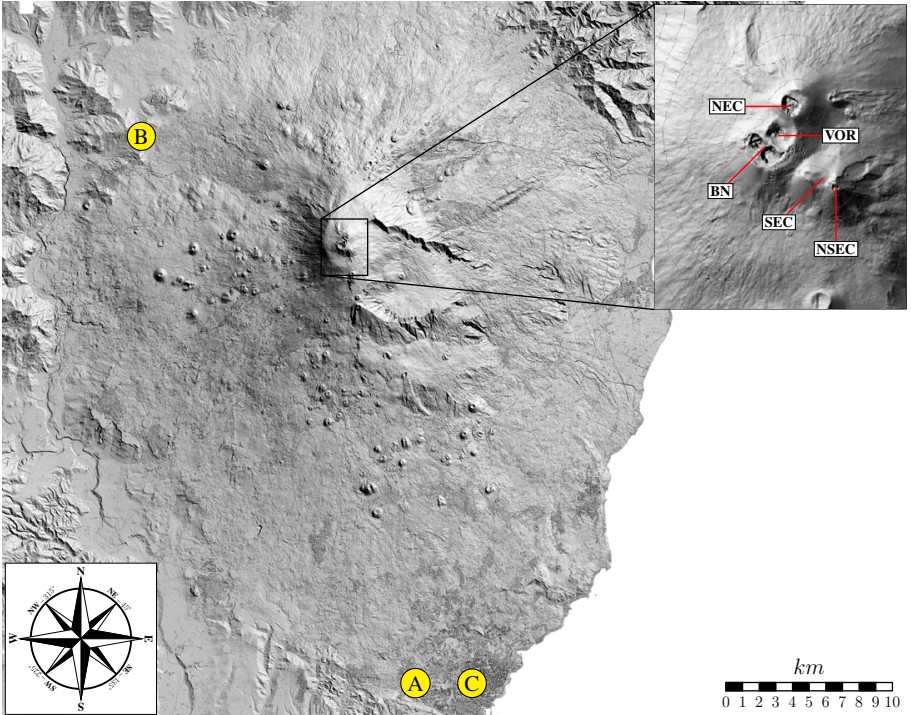

**Figure 1.** Map of Etna volcano: (**A**,**B**) yellow circles indicate the locations of the two visible cameras; (**C**) the location of the Istituto Nazionale di Geofisica e Vulcanologia, Osservatorio Etneo (INGV-OE) monitoring room. A magnified image of the summit craters indicating the northeast crater (NEC), Voragine (VOR), Bocca Nuova (BN), southeast crater (SEC), and new southeast crater (NSEC) is also shown in the upper right corner.

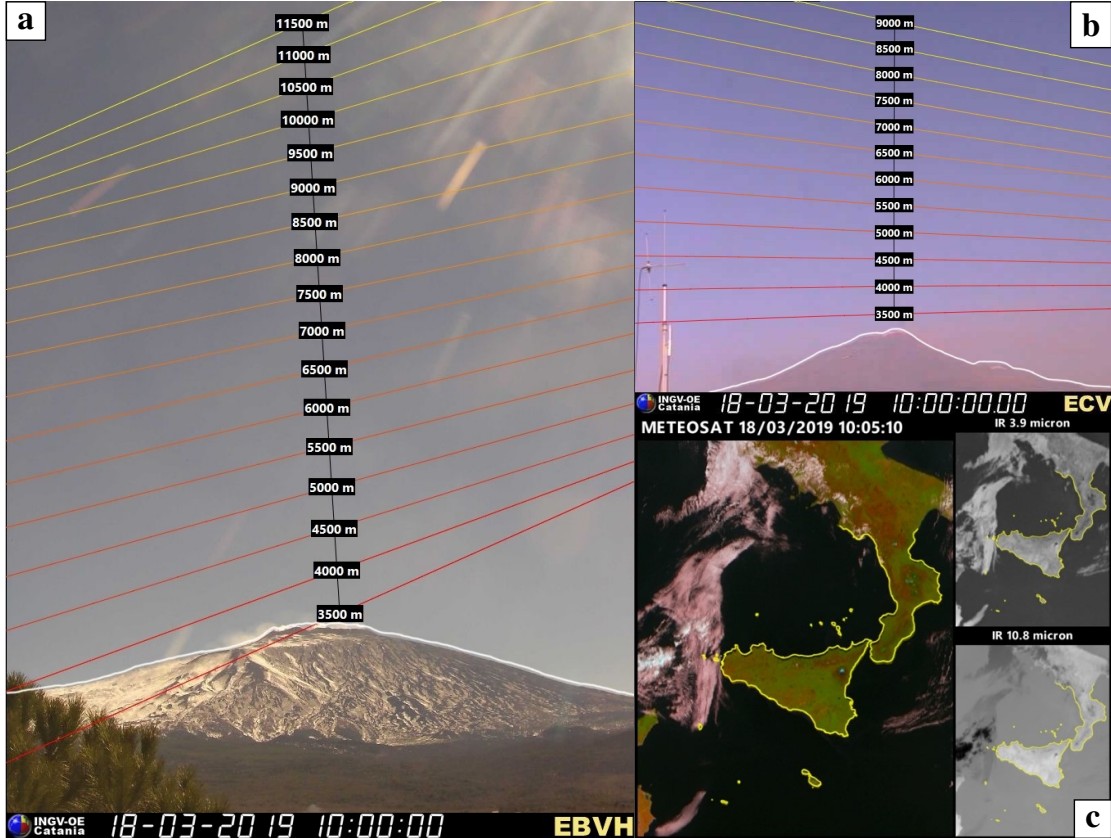

**Figure 2.** Calibrated visible images from (**a**) EBVH and (**b**) ECV cameras available during the INGV-OE surveillance and monitoring activities; (**c**) spinning enhanced visible and infrared imager (SEVIRI) image available at the INGV-OE 24/7 monitoring room. All of these images can be analyzed by the INGV-OE volcanologists to estimate the column height after an explosive event.

## 2.2. Forecasting Structure

Simulations can be run by TCP/IP protocol following three different queues: (i) emergency, (ii) daily, and (iii) research queues. The first one allows the volcanologist to run simulations during the hours after a volcanic crisis with the highest priority. The second queue has a normal priority and is automatically processed with results accessible in the operative web portal of the INGV-OE 24/7 monitoring room and to the DPC [20]. Finally, simulations can also be run for research purposes with low priority. Each request is sent to the "ash service" module, which schedules each simulation using different priority levels. The module continuously monitors the state of the simulations and processes the queue, sending a notification when the simulation starts and ends. Moreover, the use of a web socket server within the ash service allows scheduling and checking of the state of the simulations through a remote control. The database is composed of: (i) the coordinates of the volcanic vent; (ii) the reference eruptive scenarios; (iii) the model; (iv) the map module (integration domain, output resolution, and projection); and (v) the start and end times of the explosive event. The post processing analysis includes the tephra deposit or concentration in the atmosphere through the "output" module, which produces maps that are published in web portals and specific folders. Figure 3 shows the main structure of the new system.

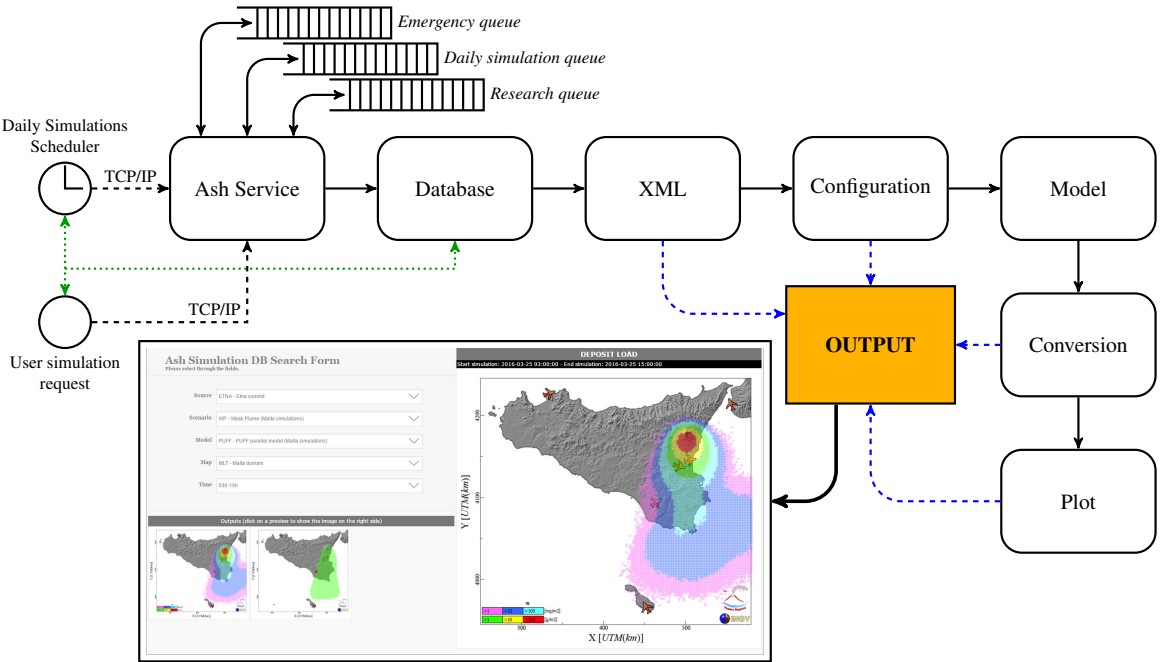

**Figure 3.** Flow chart of the new forecasting system.

## 3. Results

### 3.1. Scenario Identification

The scenario (WPS versus SPS) is identified using the scaling parameter [41], defined as Π:

$$\Pi = \frac{\overline{N}H}{1.8\, v}\left(\frac{\alpha}{\beta}\right)^2 \tag{1}$$

where H is the maximum column height above the crater (m), $v$ is wind speed averaged over the column height (m/s), $\alpha$ is the radial air entrainment coefficient, $\beta$ is the wind entrainment coefficient, and N is the buoyancy frequency (s$^{-1}$) averaged across the column height, which quantifies the stratification of the atmosphere. This is evaluated by $\overline{N} = \frac{1}{H}\int_0^H N^2(z)dx$ with $N^2(z) \sim \frac{g}{T}\frac{dT}{dz}$, where $g$ is the gravity (m/s$^2$) and $T$ is the temperature (°C). Following the methodology suggested by [42], volcanic plumes are bent over (i.e., weak plume) if Π < 0.1 or rise vertically (strong plume) if Π > 0.5, while plumes having values between 0.1 and 0.5 are considered transitional based on column shape using a 1D plume model. In our system, WPS include eruptions forming weak plumes, while SPS eruptions form transitional and strong plumes. The maximum column height is estimated during an eruption using visible cameras or satellite images. Meteorological data are obtained daily with a time step of 3 h from the weather forecasts of the hydrometeorological service of ARPA (Agenzia Regionale per la Protezione dell'Ambiente) in Emilia Romagna. The weather forecasts cover an area rotated spanning from 1 to 7.25 E and from −23 to 17.65 N with respect to the equator, and which has 22 isobaric levels [20]. Entrainment coefficients are fixed using the best values from the literature (α = 0.1 and β = 0.5) [41]. Table 1 shows a list of events where H may be estimated by satellite or camera, while the plume type (weak, transitional, or strong) may be seen through visible cameras. The average wind speed between the summit crater and H is also reported. Estimates of Π in Table 2 show that all events can be classified as weak, with some at the limit of being transitional. No volcanic events between 2011 and 2015 generated strong plumes (i.e., no plume generated significant upwind tephra sedimentation). We also evaluated the angle (in degrees) between a vertical line above the crater and the line connecting the crater and the position of the darkest pixel in the satellite image. For this analysis, we took into account only those eruptive events in which the difference between H

obtained by cameras and H obtained by satellites was less than 10% and eruptive events in which H was retrieved by satellite (e.g., during the night). Results showed that weak plumes are clearly identified when angles are greater than 65°. Results of this classification are reported in Table 2 and indicated in Figure 4 on the ECV images for comparison. For the last event (4 December 2015 ) the column height is not evaluated during the climax.

**Table 2.** Parameters of selected lava-fountain-fed plumes between 2011 and 2015: maximum column height (km) obtained using satellite (Hs) and ECV camera (Hc) sensors, time obtained by satellite, wind speed (m/s), angle (°), Π parameter (see Formula (1)), type of plume and maximum mass eruption rate (Max MER) (kg/s) are also evaluated from the column height [41].

| Eruption | Time | Hs | Hc | Wind Speed | Angle | Π | Type of Plume | Max MER |
|---|---|---|---|---|---|---|---|---|
| 10 April 2011 | 11:45 | 6.7 | 9.5 | 10.5 | 68.15 | 0.15 | Transitional plume | $1.7 \times 10^5$ |
| 12 May 2011 | 02:10 | 5.2 | | 17.2 | 63.77 | 0.06 | Weak plume | $1.7 \times 10^4$ |
| 9 July 2011 | 14:45 | 9.6 | | 8.2 | 33.45 | 0.24 | Transitional plume | $1.7 \times 10^5$ |
| 25 July 2011 | 05:10 | 5.7 | 5.3 | 18.9 | 80.10 | 0.04 | Weak plume | $1.5 \times 10^4$ |
| 12 August 2011 | 09:45 | 8.2 | 9.5 | 11.0 | 60.14 | 0.21 | Transitional plume | $1.4 \times 10^5$ |
| 20 August 2011 | 07:35 | 11.2 | | 9.5 | 48.46 | 0.37 | Transitional plume | $4.7 \times 10^5$ |
| 29 August 2011 | 04:45 | 9.6 | | 8.6 | 26.82 | 0.32 | Transitional plume | $1.3 \times 10^5$ |
| 8 September 2011 | 08:35 | 11 | | 11.4 | 57.16 | 0.36 | Transitional plume | $3.5 \times 10^5$ |
| 5 January 2012 | 07:00 | 16.2 | | 6.1 | 46.54 | 0.40 | Transitional plume | $3.6 \times 10^6$ |
| 9 February 2012 | 03:15 | 8.8 | | 6.0 | 51.55 | 0.38 | Transitional plume | $4.4 \times 10^4$ |
| 28 February 2013 | 10:35 | 9.3 | 8.0 | 23.9 | 66.48 | 0.17 | Transitional plume | $1.7 \times 10^5$ |
| 3 April 2013 | 14:25 | 5.8 | 8.2 | 27.7 | 76.79 | 0.04 | Weak plume | $2.4 \times 10^5$ |
| 12 April 2013 | 11:50 | 7.3 | 7.0 | 23.5 | 67.45 | 0.09 | Weak plume | $5.2 \times 10^4$ |
| 27 April 2013 | 18:10 | 5.2 | 7.8 | 16.8 | 72.25 | 0.04 | Weak plume | $1.9 \times 10^5$ |
| 23 November 2013 | 10:05 | 11.1 | | 43.2 | 65.56 | 0.06 | Weak plume | $1.4 \times 10^6$ |
| 4 December 2015 | 09:15 | 10.9 | | 15.7 | 41.29 | 0.35 | Transitional plume | $2.4 \times 10^5$ |

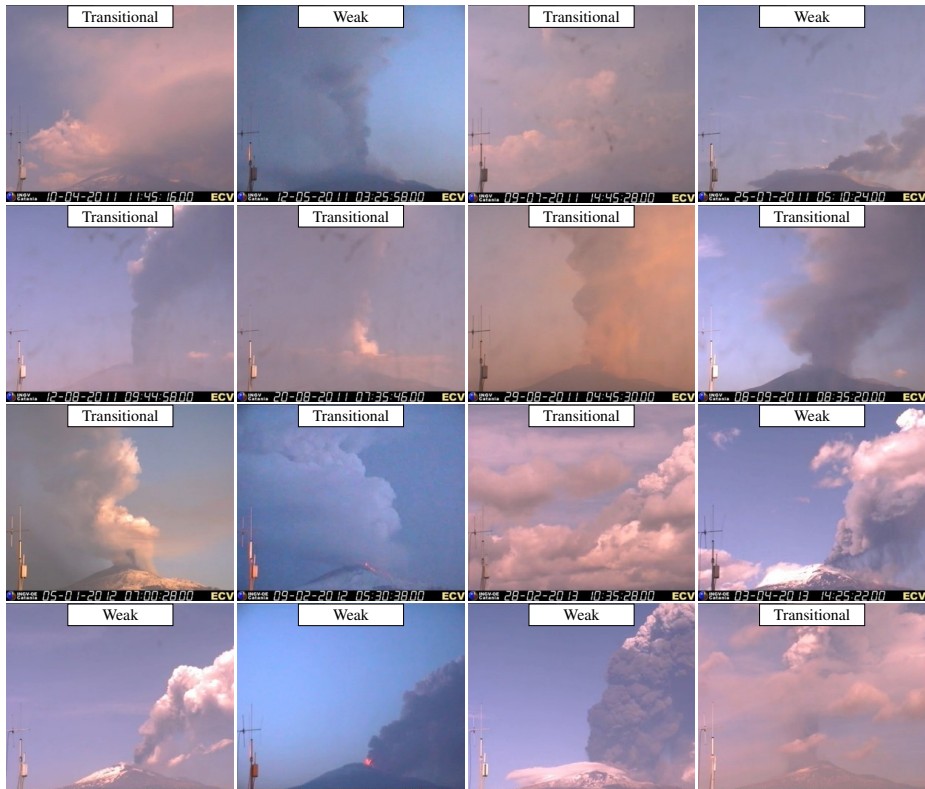

**Figure 4.** ECV images of eruption plumes generated by the lava fountains reported in Table 2, at the time the column height has the highest value retrieved by satellite. For the 12 May 2011, and 9 February 2012, the height was reached during the night and we show the ECV images closer to the climax phase of each event. The classification based on Π is also indicated.

### 3.2. Near-Real-Time Volcanic Ash Simulations

#### 3.2.1. Determination of Mass Eruption Rate from Column Height

Mass eruption rate (MER) is estimated by inverting the observed height using the 1-D plume model of Degruyter and Bonadonna [41]. Similar models are commonly used, as they can capture the first-order physics of a volcanic plume rising in the atmosphere, while remaining computationally efficient [43]. The 1-D plume model assumes that: (i) the plume is in a steady-state, (ii) the solid particle and gas phase in the plume are well mixed, such that they have the same bulk velocity and temperature (sometimes referred to as the dusty gas or pseudo-gas approximation), (iii) differences in pressure between the plume and the atmosphere are negligible, (iv) the velocity and temperature distributions through a cross-section of the plume follow a top-hat profile, which remains self-similar along the plume trajectory, (v) the radial air entrainment is $\alpha = 0.1$, and (vi) the wind entrainment $\beta = 0.5$. The governing equations are the conservation of mass, momentum, and energy, which are described in [41]. The observed meteorological conditions (atmospheric temperature and wind speed) are used as boundary conditions in the model. The wind direction instead has a negligible effect. A number of model constants need to be defined, as shown in Table 3. A single model calculation is then produced by defining four initial conditions: (i) the MER (varied), (ii) the exit velocity, (iii) the initial gas-pyroclast mixture temperature, and (iv) the initial gas mass fraction [44]. The MER of the ongoing eruption is then determined by varying the MER in the model using a bisection method, such that the resulting model height matches the observed plume height above the crater. Following [45], the model height $H_m$ is adjusted to account for bending over of the plume in the case of a weak plume:

$$H_m = \min(H_{c0}, H_c + r) \qquad (2)$$

where $H_{c0}$ is the centerline height in the case of no wind, $H_c$ is the centerline height, and $r$ is the radius of the plume. All these values are evaluated at the point where the vertical plume velocity becomes zero.

**Table 3.** List of model parameters used to evaluate the MER from the column height.

| Model Parameter | Values |
|---|---|
| gravitational acceleration (m/s$^2$) | 9.8 |
| specific heat of air at constant pressure (J/kg/K) | 998 |
| specific heat of water vapor at constant pressure (J/kg/K) | 1952 |
| specific heat of solid pyroclasts (J/kg/K) | 1250 |
| gas constant of air (J/kg/K) | 287 |
| particle density (kg/m$^3$) | 1000 |
| maximum non dimensional height (Morton et al. 1956) | 2.8 |
| radial entrainment constant | 0.1 |
| wind entrainment constant | 0.5 |
| vent height (m) | 3260 |
| exit velocity (m/s) | 100 |
| exit temperature (K) | 1400 |
| exit water fraction | 0.03 |

#### 3.2.2. Maps of Tephra Deposits

The model TEPHRA2 [5] is used to simulate isomass maps in NRT. Figure 5 shows the software interface used by the volcanologist to simulate the tephra deposit soon after an eruptive event. The three obligatory variables are the "start" and "end" of the eruption (i.e., the duration of the explosive event) and the plume height in meters above sea level (a.s.l). The volcanologist can decide to use meteorological data from the weather forecast [20] set by default or different meteorological observations (e.g., radio sounding balloons). The total grain-size distribution (TGSD) used is that of the 2002–2003 Etna eruption [21]. However, a list of the column height and the main parameters of

lognormal distribution (in phi, where phi is the $-\log_2 d$, where d is the particle diameter in mm) of TGSDs for Etna's past eruptive events [46] obtained from published papers [8,21–23,36,47–49] is also available to the volcanologist and is shown in Table 4.

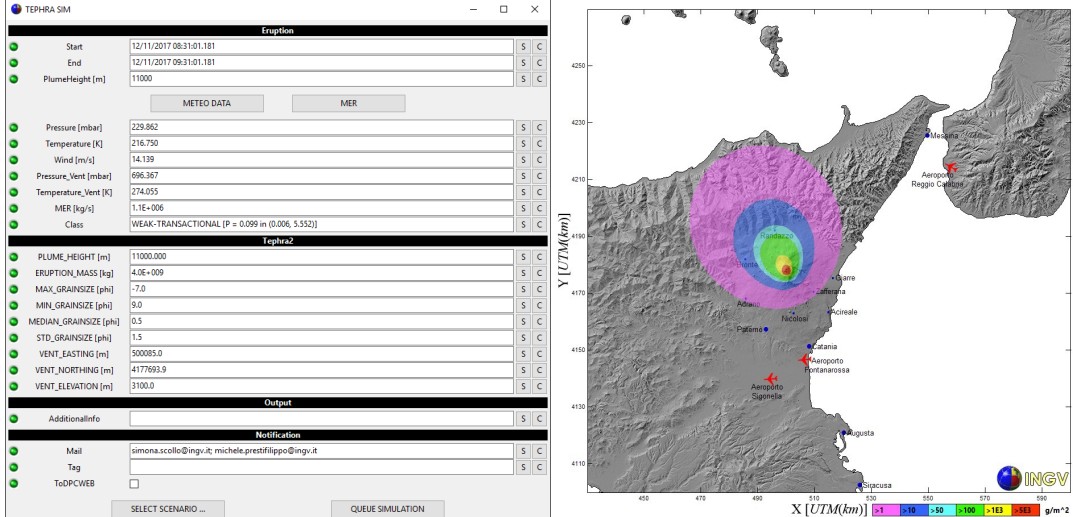

**Figure 5.** On the left is the graphical user interface used by the volcanologist and on the right is an example of the computed isomass of the tephra deposit produced by the new operational system with TEPHRA2.

**Table 4.** Column height (H) above the vent and the total grain-size distribution (TGSD) of eruptive scenarios obtained from [46].

| Eruptive Scenarios | H (km) | μ (phi) | σ (phi) |
|---|---|---|---|
| 2001 Etna eruption | 2.5 | 2.175 | 0.916 |
| 2002–2003 Etna eruption | 3.25 | 0.522 | 1.809 |
| 24 November 2006 | 0.8 | 1.714 | 0.633 |
| 4–5 September 2007 | 2 | −0.031 | 1.328 |
| 11–12 January 2011 | 7 | −1.879 | 1.782 |

The crater is centered at the new southeast crater (NSEC), but the volcanologist may change the location of the crater or lateral vent (see the graphical user interface on the left of Figure 5), because this parameter can have an effect on tephra deposit forecasts. Moreover, the volcanologist may also add any additional information before finally sending the results to the DPC. Figure 5 shows also an example of the tephra deposit isomass map obtained with the TEPHRA2 model already validated on other Etna eruptions (right of image) [23,50].

3.2.3. Application to Two Past Lava-Fountain-Fed Plumes

The system was tested on two well-studied Etna eruptions: the eruptions on 12 August 2011 [32], and on 23 November 2013, which may be considered as two end members of Etna explosive activity [8]. The event on 12 August 2011, was preceded at the NSEC by the increase of volcanic tremor and intensification of Strombolian activity, with ash emissions between 05:30 and 07:30 (hereafter, all times are in UTC). After this time, the explosions became more intense, with the generation of a lava flow occurring at 07:50 that was directed toward the Valle del Bove; at 8:30, the activity increased and a lava fountain began and remained sustained until 10:30, with a maximum column height of 9.5 ± 0.5 km a.s.l. The activity decreased and lasted until 11:00; volcanic particles were dispersed toward the SE direction, affecting the towns of Zafferana and Milo, and reaching the Ionian coast between Giarre and Acireale. The event on 23 November 2013, at the NSEC began on the afternoon of the previous day with Strombolian activity that intensified at 07:00 on November 23. A powerful lava fountain formed

at 09:30; the paroxysmal phase lasted until 10:20, and then the intensity began to decrease until 11:15; a plume about 11 km a.s.l. spread tephra to the NE and ENE, and fallout of larger clasts caused damage to vehicles and roads in the Rifugio Citelli touristic area [8], about 5 km from the NSEC. This event was well-observed with different remote sensing techniques that allowed estimate of the main ESPs [39].

For both events, we have monitored the variation of column height by camera and satellite with 5 min intervals (Figure 6). Comparison between the measurements detected by the two different sensors shows a good agreement mainly for the phases of maximum column height. This could be caused by the fact that at the start and end of the activity the volcanic plume is not optically thick enough, and larger errors in the satellite observations are expected. as well as during the first part of the climactic phase of the eruption, when the plume is still very dilute. As a result, the volcanologist should use, when possible, H obtained from calibrated cameras.

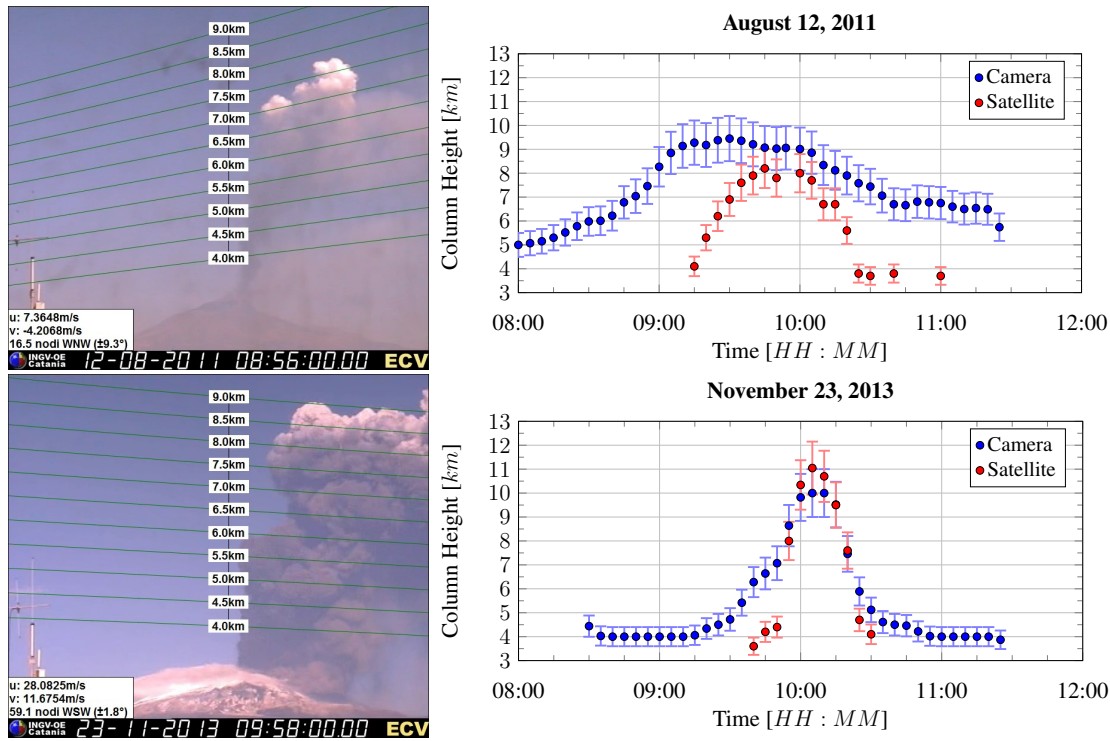

**Figure 6.** Images of the ECV calibrated camera and variation of the column height obtained by cameras (blue points) and satellites (red points) for the events on 12 August 2011, and 23 November 2013. Error bars are set to 10% of the values for both systems.

The estimation of H at different time steps (1 min, 5 min, etc.) enables us to evaluate the MER (and total erupted mass (TEM)) for each time step following the approach described in Section 3.2.1. From an operational point of view, several tests using H obtained by the visible calibrated camera were carried out to find the best procedure that should be applied during an eruption. First, the variation of H evaluated from two different operators is less than 2% for both events. The TEM using H values with 1 min intervals was compared to TEM estimates obtained using 5, 10, and 30 min intervals. The differences are less than 20% if 5 or 10 min time steps are used. Intervals of 30 min or longer should be discarded, due to the marked unsteady nature of these eruptions, with large variations of MER with time. Consequently, we suggest that the volcanologist should: (i) evaluate H at 5 min intervals (best tradeoff between precision and workload); (ii) estimate the mean MER from the time window during which H reaches the highest values (we found errors less than 15% for both events). This temporal interval also avoids the high uncertainty in the satellite column height previously described. Figure 7 shows that we can use the derivative of the cumulative of TEM to estimate the time in which H has the highest values. A simple application is available to the volcanologist and is able to estimate it

semi-automatically. It is notable that greater errors are expected (44% and 96%, respectively) when the mean value of H is considered for the entire eruption. Figure 7 also shows satellite images retrieved at 10:00 GMT and the tephra deposits obtained for 12 August 2011, and 23 November 2013, following our approach. Results for 23 November 2013, are in good agreement with the isomass map reported in [51], as shown in Figure 8. The computed mass loading is in fact inside one-fifth and five times the observed thresholds that are used for modelling validation purposes [27].

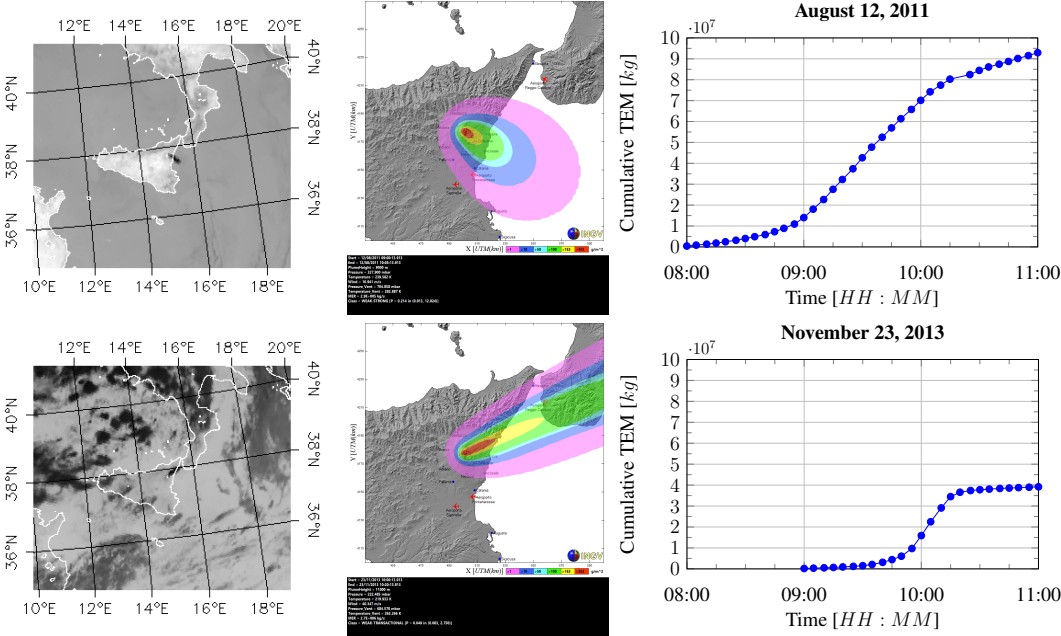

**Figure 7.** On the right are satellite images retrieving volcanic plumes at 10:00 GMT. In the middle are results of the computed tephra deposit obtained by the new system, On the left are the TEM cumulative curves for 12 August 2011, and 23 November 2013, respectively.

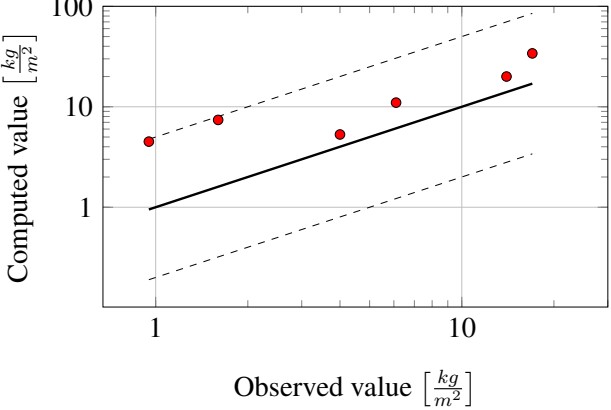

**Figure 8.** Comparison between the punctual tephra mass loading measured on the ground and reported in [51], and values computed with TEPHRA2 for the event on 23 November 2013. Dashed lines indicate over and under estimation of one-fifth and five times the observed values.

### 3.3. Impact of Large Clasts (≥ 5 cm)

Sedimentation of large clasts is computed with the model described in [52] and applied for the first time at Etna [53]. The model has an approach similar to that presented in [54]. The same 1-D plume model as in Section 3.2.1. describes the plume physics and trajectory. The plume cross-sectional velocity top-hat profiles are converted to Gaussian profiles. Clast support envelopes within the plume are then calculated by equating the cross-sectional plume vertical velocity to the respective clast

terminal velocity. From each envelope, volcanic particles are released from randomly defined heights and their trajectory through the atmosphere is calculated until they reach the ground. The particle sedimentation is obtained numerically from the equation of motion, which takes into account advection by wind, drag, and gravitational forces. The number of particles used in our simulations (500 clasts) is in agreement with previous comparisons between the model results and field data already validated by [53] for the event on 23 November 2013. Two end member reference scenarios are simulated daily based on two well-studied events: (i) the event on 12 January 2011 [36], considered as a low intensity event; and ii) the event on 23 November 2013 [8], which is one of the most explosive events that has occurred in recent years. These scenarios are run every three hours following the approach of [20]. Table 5 shows the main model inputs and Figure 9 shows the results of the automatically generated simulation. The crater location for those events is the NSEC, which can be changed.

**Table 5.** List of model parameters used to run large clast simulations [53].

| Model Parameter | Values |
|---|---|
| diameter of volcanic vent (m) | 50 |
| scenario with high MER | 23 November 2013 |
| scenario with low MER | 12 January 2011 |
| particle density (kg/m$^3$) | 865 |
| plume velocity (m/s) | 100–150 |
| diameter of clasts (cm) | 5 |
| vent location | NSEC |

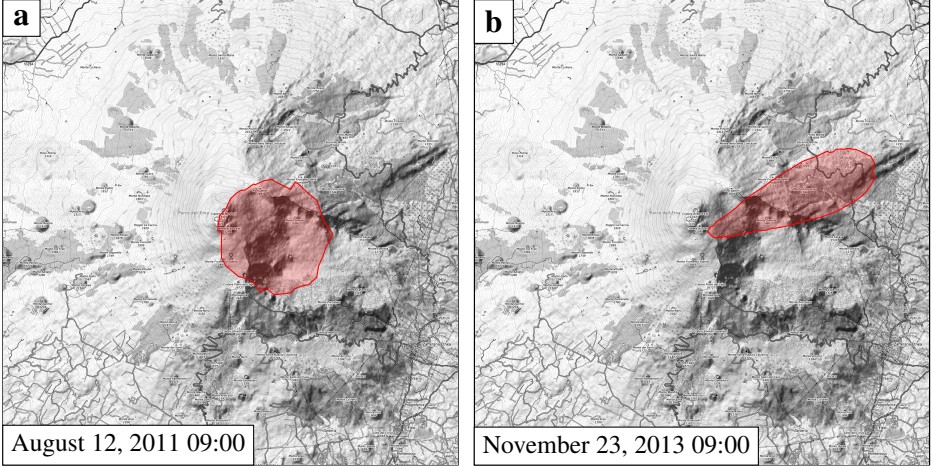

**Figure 9.** Map of large clast deposit for the eruptive scenarios based on 12 January 2011 (**a**), and 23 November 2013 (**b**).

## 4. Discussion

In this work, we show a new operational system implemented in the 24/7 Etna volcano surveillance that is able to model plume dispersal and fallout in NRT using volcanological observations collected in NRT. Our volcanological observations are based on the analysis of images from low-cost visible and calibrated cameras and satellites. The system also computes the isopleth maps using two end member eruption scenarios every three hours and helps the DPC to identify regions experiencing sedimentation of large clasts (≥ 5 cm).

### 4.1. Column Height Estimation

In our system, column height can be estimated using images from visible calibrated cameras and satellite images. Although visible calibrated cameras look promising, column heights are only estimated during the day. Analyzing past eruptions between 2011 and 2015, about 50% of events

occurred late in the day in dark sky conditions and were not retrieved by camera. Excluding those events, the visible calibrated camera was able to retrieve almost 80% of events. It is highlighted that for past events, the column height crossed over the field of view of the ECV camera and was not retrieved during the climax. However, the new camera has a higher field of view, allowing detection of column heights rising up to about 15 km a.s.l. Moreover, new studies on the use of thermal cameras aimed at retrieval of volcanic plumes could increase the capacity of plume detection using ground-based systems [55,56]. It is worth noting that although the analysis could have greater errors for volcanic plumes with insufficient optically thickness, the probability of success in the H estimations by satellite is about 65% [38].

### 4.2. Total Erupted Mass and Mass Eruption Rate

The TEM is one of the most important ESPs [50]. The TEM for past eruptions has typically been obtained by field data using curve-fitting techniques over the isopach curves of the deposit [57], or from the solution of an inverse problem [50]. The mean MER can then be estimated by dividing the TEM by the duration of the eruption. In contrast, the MER determined from plume height [58] is typically associated with the maximum plume height, and therefore represents the maximum MER. In the proposed procedure, the MER is instead averaged over a restricted time window around its maximum value, so representing a mean "maximum" value representative of the part of eruption during which most of the volume was erupted. Additionally, in some formulations for the determination of MER from plume height, the wind condition is not included explicitly [58]. In our system, we use a modified version of the model in [41], which in general describes the weak-plume behavior well and may also be applied when volcanic plumes are affected by strong winds. In this case, the calculation of MER depends on the time resolution at which plume height is determined.

### 4.3. Total Grain-Size Distribution

The TGSD is very difficult to estimate in NRT. The main properties of tephra particles (e.g., size, shape, and density) are strictly related to magma rheology, composition, vesiculation and fragmentation dynamics, and eruptive style, and they may vary for different volcanoes, different eruptions from the same volcano, and even for different phases of the same eruption (e.g., the 2001 and 2002–2003 Etna eruptions). The TGSD is usually evaluated from the integration over the dispersal area of the grain-size distribution of samples collected in the field, and is representative of the whole deposit at each site [59], even though some attempts to complement TGSD with satellite retrievals [60] or to determine TGSD with weather radar also exist [61,62]. The main complication is that remote sensing techniques cannot detect all particle sizes that are typical for volcanic plumes; in fact, these techniques are limited to specific windows of operation (e.g., wavelength) [28]. Therefore, the determination of TGSD in NRT is still challenging, and the use of a fixed TGSD associated with past reference events remains the best strategy. Consequently, uncertainty in the tephra dispersal forecast and the computed isomass maps associated with the use of TGSD of past events is expected. It is worth noting that large differences in the computed tephra deposits are expected for variation of at least 2 phi in the modal values of TGSD [63].

### 4.4. Assessment of Hazard Associated with the Sedimentation of Large Clasts

In our study, fixed scenarios are used to estimate the hazard from the fallout of large clasts. The delimitation of the area having a probability of impact of at least 1% for a 5 cm clast during a single eruptive event can be used as a hazard map [53]. As reported in [53], the evacuation of people from such an area can take several hours for both the eruptive reference scenarios selected. There is, hence, the need to give "a priori" information about the area that should be interdicted, and that during an eruption could be exposed to large-clast fallout from plume margins. Because time is critical for the emergency management during a volcanic crisis, the use of two fixed eruptive scenarios, having low and high MERs, was favored over NRT forecasting, which can take several hours to compute.

### 4.5. NRT Determination of ESPs

In the present system, the estimation of ESPs in the course of the eruption is not fully automated. The volcanologist estimates the maximum H via image analysis of satellite or camera data and the eruption duration based on TEM cumulative curves (Figure 7). Automated detection of H by cameras could further reduce the time necessary to compute the isomass maps of the associated tephra deposits. Moreover, the use of only a few instruments for plume monitoring could be critical, as under some circumstances (for example at night, when the visible cameras cannot be used) they may not work properly; as such, a multi-instrument approach (e.g., radar [64], satellite [38], and cameras [65]) is favored. However, when using multiple instruments, some discrepancies among data could exist, and the role of the operator is still essential. For all these reasons, we propose a system similar to the earthquakes alert that works well at INGV-OE. Within a time window of 5 min after the end of the eruptive event or after 3 h from the eruption onset for a long-lasting event, an automatic system discriminates the eruption scenarios based on the maximum value of H. This value is inserted in the Volcano Observatory Notices for Aviation (VONA) message. Given H and meteorological forecasts, the new NRT system automatically sends the most reliable forecasts to the DPC from the model results computed with fixed scenarios. Then, within the first 3 h after the end of the eruption or after three hours from the onset of the explosive event (if there is a long-lasting activity), the volcanologist will check the reliability of H values obtained by the instruments (in our case from cameras and satellites) and will give a more accurate forecast. In the future, operators who work at the INGV-OE 24/7 operational room could run the model and send the computed tephra isomass map to the DPC. The results of simulations should be easily accessible to the authorities and decision makers in order to reduce the impact that explosive eruptions may have on the surrounding population.

### 4.6. Advantage of the Proposed System

Our system has several advantages. First of all, the volcanologists and DPC may observe and quantify column height during an eruption in NRT using two different remote sensing systems (e.g., visual HD camera and satellite images). From those estimations, the volcanologist can appreciate the potential impact that the eruption will have on the population. Moreover, most of the existing NRT systems are mainly based on satellite observations, except for the Icelandic Meteorological Office (IMO), which also receives column height values from radars. Calibrated visible cameras applied to volcano surveillance can provide very useful information, and being low cost, could spread worldwide with relative ease. The computation of isomass maps gives the DPC a more reliable computed map of the tephra fallout. In this case, the DPC can carry out efficient actions to mitigate the impact associated with tephra fallout. Moreover, the isopleth map modelled every three hours could reduce the impact associated with fallout of large clasts at Etna, considering that the evacuation on foot to a safe area was estimated to take [51] almost 4 h. Finally, the selection of the best eruption scenarios run daily by INGV-OE every three hours gives the DPC fast information on the areas that have high probability of being affected by tephra fallout.

## 5. Conclusions

In this study, we explore the capability of computing the isomass maps of tephra deposit and the isopleth maps of clasts $\geq 5$ cm falling from plume margins using observations of volcanic activity in NRT. We use low-cost, visible, calibrated cameras and satellite images to estimate the column height variation and the time period in which the column height reaches the maximum values. Moreover, the proposed system is also able to automatically discriminate the most reliable eruptive scenarios among weak, transitional, and strong plume scenarios, and the region of fallout deposit of large clasts ($\geq 5$ cm) that could cause injuries to people and damage infrastructures in proximal areas.

This platform is already implemented in the INGV-OE 24/7 Etna volcano surveillance. The system is semiautomatic because it requires the input of an operator (volcanologist in our case) to check the

column height obtained from satellite and calibrated images and to run simulations. However, we are confident that the system will become fully automatic in the near future.

Results of our system have been validated with two Etna eruptions, providing good agreement. Given the reliability of the proposed approach, the authors believe that it can also be applied to volcano observatories worldwide, in particular in those volcano observatories that have to provide information to civil protection authorities in NRT.

**Author Contributions:** Conceptualization: S.S. and M.P.; methodology: M.P., S.S., C.B., S.C., and W.D.; software: M.P., E.R., W.D., and S.S.; validation: S.S, M.P., G.C., R.C., S.C., and L.M.; data curation: E.B., E.P., M.S., M.M., and M.P.; writing—original draft preparation: S.S.; review and editing: C.B., R.C., S.C., W.D., L.M., E.P.; M.P., E.R., M.S., and S.S.; visualization, C.C. and M.P.

**Funding:** The work has benefited from funding provided by the Italian Presidenza del Consiglio dei Ministri—Dipartimento della Protezione Civile (DPC). This paper does not necessarily represent DPC's official opinion and policy. This project has also received funding from Excellent Young Scientists 2015 Istituto Nazionale di Geofisica e Vulcanologia call and from the European Union's Horizon 2020 research and innovation program under grant agreement No 731070 (Eurovolc project).

**Acknowledgments:** The authors are grateful to Mauro Rosi and Stefano Ciolli for supporting the improvement of the volcanic ash monitoring and forecasting system at Etna. Eugenio Privitera, director of INGV-OE, and Stefano Branca, head of the INGV-OE volcanologist department, are thanked for their positive feedback when making the new system inside the 24/7 surveillance department of Etna volcano. We thank Gianluca Carà, who is in charge of the administration procedures of the INGV-OE projects. Augusto Neri, Sonia Calvari, Giovanni Macedonio, Daniele Andronico, Mauro Coltelli are also thanked.

**Conflicts of Interest:** The authors declare no conflict of interest. The funders had no role in the design of the study; in the collection, analyses, or interpretation of data; in the writing of the manuscript, or in the decision to publish the results.

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
