# Peer review of "Near-Real-Time Tephra Fallout Assessment at Mt. Etna, Italy"

_remotesensing, doi:10.3390/rs11242987_

Round 1

Reviewer 1 Report

I think the revised manuscript is improved. I think it is worth to publish this paper in remote sensing. Thanks for the author's efforts. 

Author Response

We thanks the reviewer for her/his comment.

Reviewer 2 Report

I am satisfied with the revisions in response to my comments. The table you provided would be useful for those who might be interesting in adopting such a system. Thanks for the effort that you made! Now I only suggest improving the introduction by making a separate paragraph detailing the objectives of the study/project.

Author Response

We thanks the reviewer for her/his comments and suggestions. We divided the introduction in two separate sections: 1.1. Worldwide operational monitoring and forecasting of tephra dispersal and fallout and 1.2. Main Objectifies of the new upgraded system at INGV-OE that was also improved. Moreover, we made minor improvements in English language and style. 

Reviewer 3 Report

The revisions on the manuscript have polished the presentation and does the novel techniques presented therein justice. It is a valuable contribution to the special volume it was prepared for. 

Author Response

We thank the reviewer for her/his comment. Minor spell check was made. 

Reviewer 4 Report

All the comments/answers to the previous 4 reviews have been addressed.

I don't have anymore comments to add to this manuscript.

Author Response

We thank the reviewer for her/his comment. Minor improvements in the English language and style was also made. 

This manuscript is a resubmission of an earlier submission. The following is a list of the peer review reports and author responses from that submission.

Round 1

Reviewer 1 Report

Please find my review of the manuscript “Near real-time tephra fallout assessment at Mt. Etna, Italy” submitted to Remote sensing by Scollo et al.

This paper proposes new system of near-real time simulation on tephra fallout at Mt. Etna. The system monitors volcanic plume by two HD visible cameras and satellite, and forecasts isomass map of tephra by a tephra fall simulator with observed plume height in near real time. The author was applied the system to past eruptive events at Mt. Etna, showing maps of isomass and distribution of large clast (> 5 cm).

This paper is well organized, and clearly presents strategy of near real-time assessment of tephra fallout by the system. The structures of system use well developed models on estimation of mass eruption rate (Degruyter and Bonadonna, (2012) and dispersion of tephra (TEPHRA 2) which is favored to science-based evaluation of eruptive activity. Therefore I recommend to publish this paper in Remote sensing after minor revisions. I summarized minor comments in the following.

Minor comments

Figure 1. Can you enlarge the inlet of this figure? The present size is difficult to see the location and shape of vents a little.

L219. Was wind direction used to boundary condition of the model?

L223. How do you set the exit water fraction to 0.03? Is there reference?

L270. Although the author suggests “there is only good agreement during the climax”, there seems to be the difference of peaking times of column height estimated from camera and satellite in 12 August 2011. So, I concern that the satellite image did not capture even the timing of climax.

Figure 7&8. I am wondering why the author did not compare the model outputs with the actual maps of isomass and distribution of large clast which can be obtained the field survey. If there are the maps, I hope the author compared it with the model outputs. The comparison strengths the usefulness of the system.

Reviewer 2 Report

Comments on remotesensing-624036

In general, I believe that this piece of work illustrates a good example of the remote observations of tephra fall at Etna, one of the most active volcanoes in the world. This work as well as the system described in the paper would be useful in the practice of hazard monitoring and forecasting; however, the innovation seems limited to me from the perspective of remote sensing. As such, I consider this paper more suitable for Sensor (also by MDPI), rather than Remote Sensing.

Specific comments:

Since you claim that this system can be easily adapted for worldwide application, I suggest adding a figure detailing the component of the monitoring system. Try to include the sensors used and, if possible, the cost for setting up the system. What are the (potential) disadvantages of this system? And what are the solutions? These would be important for those who might be interested in such systems. Figure 6: blue points for camera observations and red for satellite observations. The manuscript might need check for grammatical errors and spelling (e.g. Lien 429 Dr.ssa)

Reviewer 3 Report

This paper detailing a new semi-automated monitoring workflow for explosive eruptions at Mt. Etna is well written and of interest to a wide audience. 

I have only a few comments/suggestion to help improve the flow of the methods section, and 2 questions for clarity within the text. These suggestions are very minor and easily remedied. 

The method section has a few elements out of order, that if structured in a more consistent fashion would help the readers. For example: the error estimates of the visual plume height occurs within the section on satellite based plume heights. 

When does the vent location get identified through this work flow? Does it impact the height estimate? or only numerical simulations? This just needs to be stated in the appropriate section for clarity. 

With the abundance of data available for Etna, can the authors estimate how many events were not detectable by cameras for H measurements or by satellite? I.e. the workflow cannot proceed. The paper implies this value is low, but if quantified would be very compelling for the success of this method.

A few notes on language/typos by line appear below: 

Line 95: Is the best selection process being designed, or the selection of the best model the goal? Unclear.

Line113: I get the principle of calibrated camera view for plume estimation, but I think the sentence can be reworded to make it clear what processing is required of the image to make these estimates.

Line 137: I suggest moving the uncertainty measurement from the visible cameras up to where it is introduced, rather than hidden here with the Satellite based uncertainty.

Line 165: Is there a mechanism in this work flow for identifying which vent is responsible? How much does this impact the simulations? To a lesser extent, does the vent impact the height estimates?

Line 183: Where are the meteorological data stations? Or is it satellite based? Just a note on the nature of the data would help build the story of this procedure.

Table 2: ) missing  in gas constant of air line

Table 4 and line 315- how does vent location come into this?

Line 336: “in general describes the weak plume behavior well”

Line 350: any attempts to quantify this uncertainty? Or range of uncertainty?

Reviewer 4 Report

General comment:

Based on 2 case studies, this paper shows the new NRT system of volcano monitoring related to Tephra fallout (Mt Etna) implemented at INGVINGV-OE. This is well presented. Nice to read the state of the art of previous study related to a such NRT service. I recommend a publication after minor review (typo + some specification of the methodology).

Looking forward for seeing the results this NRT service will obtain in the future.

Comments and typo:

l46: thx to correct this "within t 5–6 km"

l96-97: eventually you can replace "near-real time" by "NRT"

l99-100: eventually you can replace "near-real time" by "NRT"

l108: in "The Focal Length is 3.5 - 10 mm and the Aperture", the use of -uppercase letters is not essential, lowercase letter seems good to me. Thanks to mention that the setting of the focal length can range from 3.5 to 10 mm. It will be good to know the settings used by the 2 cameras (focal lengths and apertures used, i.e. is it wide or tele?, the horizontal, vertical diagonal fields of view)

l109: replace "F 2.7(tele)" by "F 2.7 (tele)".

l113: thanks to specify the acronym of DEM

l130: maybe replace "useful" by "relevant"

l76: the parameters/coefficients of Eq.1 are well specified in this paragraph, however I don't see the estimate of the buoyancy frequency (N) used. Thanks to specify it.

l190-192: Thanks to rephrase / clarify this methodology

l190: it might be good to mention that no strong event are presented in Table 1.

l200: in Table1, the tick used for weak and Trans are (X and x, respectively). Thanks to unify the style of tick if justifies.

l231: replace "^2" by "2" and "^3" by "3"

l240-241 or l242-244: remove "The standard deviation 243 of TGSD is fixed to 1.5"

l255: to homogeneise the wayu of writting the time, replace "05.30" and "07.30" by "05:30" and "07:30" respectively.

l258: replace "10.30" by "10:30"

l259: replace "11.00" by "11:00"

l262: replace "of 23 November" by "on 23 November"

l323: to be consistent, replace "near real time" by "near-real time"

l326: to improve the read, it might be good to replace "(≥5 cm)" by "(≥ 5 cm)"

l331: add "(MER)" after "mass eruption rate"

l364: add "(ESP)" after "eruption source parameters"

l401: to be consistent, replace "near real time" by "near-real time"

l405: to improve the read, it might be good to replace "(≥5 cm)" by "(≥ 5 cm)"